# Improving generalization by regularizing in $L^2$ function space

## Abstract

Learning rules for neural networks necessarily include some form of regularization. Most regularization techniques are conceptualized and implemented in the space of parameters. However, it is also possible to regularize in the space of functions. Here, we propose to measure networks in an $L^2$ Hilbert space, and test a learning rule that regularizes the distance a network can travel through $L^2$-space each update. This approach is inspired by the slow movement of gradient descent through parameter space as well as by the natural gradient, which can be derived from a regularization term upon functional change. The resulting learning rule, which we call Hilbert-constrained gradient descent (HCGD), is thus closely related to the natural gradient but regularizes a different and more calculable metric over the space of functions. Experiments show that the HCGD is efficient and leads to considerably better generalization.

## 1 Introduction

Large neural networks can overfit to training data, but we desire networks that instead learn general aspects that apply to new data. A learning rule can encourage generalization through regularization, which may be implicit to the learning rule or explicitly added to the cost function. Many regularization techniques introduce penalties in the space of network parameters. Canonical examples include weight decay and the constraints upon parameter movement inherent to Stochastic Gradient Descent (SGD). Ultimately, however, it is the complexity of the parameterized function that we seek to regularize, not pf the parameters themselves. A well-known example of this more direct approach is the natural gradient, which constrains how much the output distribution of a network can change due to an update.

Here, we introduce a new learning rule that, like the natural gradient, limits how much the output function can change during learning. However, we use a different and more calculable metric over function space: the expected $L^2$ norm. Since the $L^2$-space is a Hilbert space, we term the rule Hilbert-constrained gradient descent (HCGD).

The interpretation of the natural gradient as resulting from regularization competes in the literature with many other interpretations and justifications. In order to establish a foundation for the Hilbert constraint as analogous to the natural gradient, we begin by reviewing and discussing the natural gradient.

### 1.1 The natural gradient

In simplest mathematical terms, one performs the natural gradient by computing an expectation of the covariance of gradients $\mathbb{E}_{\mathbb{P}_\theta}[JJ^T]$, and then multiplying the current gradients by its inverse. This covariance matrix is known as the Fisher information metric $F$, and the natural gradient step is in the direction $F^{-1}J$. In addition to being seen as a regularizer of functional change, as we make precise below, variants of the natural gradient have appeared with no less than four justifications. These are data efficiency, minimizing a regret bound during learning, speeding optimization, and the benefits of whitened gradients. We review these disparate methods to show that they are equivalent, and to emphasize how widespread the natural gradient is for the optimization of neural networks.

Amari originally developed the natural gradient in the light of information geometry and efficiency (Amari et al. (1996); Amari (1998)). If some directions in parameter space are more informative

of the network's outputs than others, then we might wish to scale updates by each dimension's informativeness. In the terminology of Amari, this is to say that we want to find a learning rule that works when the loss geometry is not Euclidean but Riemannian (Amari (1997)). We can equivalently speak of the informativeness of new examples. If not all examples carry equal information about a distribution, then the update step should be modified to make use of highly informative examples. That is, we wish to find a Fisher-efficient algorithm (see Amari et al. (2000)). The natural gradient uses the Fisher information matrix to scale the update by parameters' informativeness.

The 'adaptive' family of learning rules derive from a framework closely related to data efficiency. The original Adagrad paper showed that the $F^{-1}J$ update reduces the bound on the regret relative to gradient descent (Duchi et al. (2011)). There, however, $F$ is referred to as the Malahanobis norm and is computed over the same examples as $J$. (Note that since the Fisher is properly taken in expectation over the output distribution, the $F$ computed over the training batch is referred to elsewhere as the 'empirical Fisher'.) The number of parameters in neural networks typically prohibits calculating and storing a full gradient covariance matrix, let alone inverting one. Adagrad is thus most commonly implemented using a diagonal approximation of the empirical Fisher. Related adaptive methods like Adadelta and RMSprop can be also be seen as employing related diagonal approximations of the empirical Fisher (Zeiler (2012)). These update rules are in wide use in neural network optimization but not commonly referred to as approximations of the natural gradient.

There is a strong connection between the natural gradient and techniques that normalize and whiten gradients. The term $F^{-1}J$, after all, simply ensures that steps are made in a parameter space that is whitened by the covariance of the gradients. Whitening the gradients thus has the effect that SGD becomes more similar to the natural gradient. Activation whitening methods are also known to speed convergence. It appears that many approaches to normalize and whiten activations or gradients have been forwarded in the literature (Raiko et al. (2012);Simard et al. (1998); Schraudolph & Sejnowski (1996); Crammer et al. (2009); Wang et al. (2013); Le Cun et al. (1991); Schraudolph (1998)). A similar effect is at play for Batch Normalization, as well (Ioffe & Szegedy (2015)). By normalizing and whitening the gradients, or by proxy, the activations, these various methods ensure that parameter space is a better proxy for function space.

A good review of the thought behind recent natural gradient approximations can be found in Martens & Grosse (2015) and Martens (2014). K-FAC is likely the most accurate scalable approximation of the natural gradient to date, reported to converge in 14x fewer iterations than SGD with momentum (and in less real time). Additionally, Pascanu and Bengio provide a clear exposition of the natural gradient as taking constant steps in the space of output distributions (Pascanu & Bengio (2013)), as measured by the Kullbeck-Leibler (KL) divergence. We present a version of this argument below but with emphasis on the interpretation as regularization.

## 1.2 THE NATURAL GRADIENT AS REGULATOR OF FUNCTIONAL CHANGE

Here we show that the natural gradient is the optimal update when one penalizes the incremental change in a network's output distribution as measured by the KL divergence.

A simple way to characterize a function is to examine the function's outputs for a set of inputs. Given an input distribution $\mathbb{X}$, we will write the distribution of the network's outputs as $\mathbb{P}_\theta$, where $\theta$ is the set of the network's parameters. Let us plan to regularize the change in the output distribution $\mathbb{P}_\theta$ throughout optimization of the parameters $\theta$. For this, we need a measure of similarity between two distributions. One such measure is the Kullbeck-Leibler (KL) divergence. The KL divergence from $\mathbb{P}_{\theta_t}$ to $\mathbb{P}_{\theta_{t+1}}$ is defined as

$$D_{KL}(\mathbb{P}_{\theta_{t+1}} \| \mathbb{P}_{\theta_t}) = \int \log \left( \frac{P_{\theta_t}(x)}{P_{\theta_{t+1}}(x)} \right) P_\theta(x) dx \tag{1}$$

$P_\theta$ is the density of $\mathbb{P}_\theta$. To ensure the output distribution changes little throughout optimization, we define a new cost function

$$C = C_0 + \lambda D_{KL}(\mathbb{P}_{\theta_{t+1}} \| \mathbb{P}_{\theta_t}) \tag{2}$$

where $C_0$ is the original cost function and $\lambda$ is a hyperparameter that controls the importance of this regularization term. Optimization would be performed with respect to the proposed update $\theta_{t+1}$.

In information theory, the KL divergence from $\mathbb{P}_{\theta_t}$ to $\mathbb{P}_{\theta_{t+1}}$ is called the relative entropy of $\mathbb{P}_{\theta_{t+1}}$ with respect to $\mathbb{P}_{\theta_t}$. We can thus equivalently speak of regularizing the change in the entropy of the

output distribution throughout learning. If we assume that the initial parameterization leads to an output distribution with maximal entropy (as would be the case if, reasonably, the inputs are treated nearly equally by the initial parameterization), and the average update decreases output entropy, then regularizing the relative entropy between optimization steps will lead to a solution with higher entropy.

Evaluating the KL divergence directly is problematic because it is infeasible to define the output density $\mathbb{P}_\theta$ everywhere. One can obtain a more calculable form by expanding $D_{KL}(\mathbb{P}_{\theta_{t+1}} \| \mathbb{P}_{\theta_t})$ around $\theta_t$ to second order with respect to $\theta$. The Hessian of the KL divergence is the Fisher information metric $F$. If J is the the Jacobian $J = (\nabla_\theta C_0)$, the Fisher metric is defined as $\mathbb{E}_{\mathbb{P}_\theta}[JJ^T]$, i.e. the expectation (over the output distribution) of the covariance matrix of the gradients. With $\Delta\theta \equiv (\theta_{t+1} - \theta_t)$, we can rewrite our regularized cost function as

$$C \approx C_0 + \frac{1}{2}\Delta\theta^T F \Delta\theta \tag{3}$$

To optimize $C$ via gradient descent we first replace $C_0$ with its first order approximation.

$$C \approx J^T \Delta\theta + \frac{\lambda}{2}\Delta\theta^T F \Delta\theta \tag{4}$$

At each evaluation, $J$ is evaluated before any step is made, and we seek the value of $\Delta\theta$ that minimizes Equation 2. By setting the derivative with respect to $\Delta\theta$ zero, we can see that this value is

$$\Delta\theta = \frac{1}{\lambda}F^{-1}J \tag{5}$$

When $\lambda = 1$ this update is exactly equal to the natural gradient. Thus, the natural gradient emerges as the optimal update when one explicitly regularizes the change in the output distribution during learning.

## 2 PROPOSED ALGORITHM

The Fisher metric is not the only way to define (and regularize) a space of functions. We propose instead to use the $L^2$ space with norm:

$$\|f\|^2 = \int_{\mathbb{X}} |f|^2 d\mu$$

Here $\mu$ is a measure and corresponds to the probability density of the input distribution $\mathbb{X}$. The $|\cdot|^2$ operator refers to the 2-norm to account for vector-valued functions. This norm leads to a notion of distance between two functions $f$ and $g$ given by

$$\|f - g\|^2 = \int_{\mathbb{X}} |f - g|^2 d\mu$$

Since $\mu$ is a density, $\int_{\mathbb{X}} d\mu = 1$, and we can write

$$\|f - g\|^2 = \mathbb{E}_{\mathbb{X}}[|f(x) - g(x)|^2]$$

We will regularize change of a network's output function as measured by this notion of distance. If a network would have been trained to adjust the parameters $\theta$ to minimize some cost $C_0$, we will instead minimize at each step $t$ a new cost given by:

$$C = C_0 + \lambda\|f_{\theta_t} - f_{\theta_t + \Delta\theta}\| \tag{6}$$

Like all regularization terms, this can also be viewed as a Langrangian that satisfies a constraint. Here, this constraint is upon gradient descent and ensures that change in $L^2$-space does not exceed some constant value.

To evaluate Equation 6, we can approximate the norm with an empirical expectation over $\mathbb{X}$.

$$C = C_0 + \lambda\Big(\frac{1}{N}\sum_{i=0}^{N} |f_{\theta_t}(x_i) - f_{\theta_t + \Delta\theta}(x_i)|^2\Big)^{1/2}$$

Here, the data $x_i$ may derive from some validation batch but must pull from the same distribution $\mathbb{X}$. This cost function imposes a penalty upon the difference between the output of the current network at time $t$ and the proposed network at $t + 1$. This cost implies a learning rule, which we call Hilbert-constrained gradient descent (HCGD).

We can write an update rule to minimize Equation 6 that is a modification of gradient descent. Our implementation, displayed as Algorithm 1, takes some lessons from the natural gradient. Just as the natural gradient is the optimal solution to Equation 4 at each step, here we seek the optimal solution to Equation 6. Thus we seek to converge to a $\Delta\theta'$ at each update step, where

$$\Delta\theta' = \underset{\Delta\theta}{\mathrm{argmin}} \left( J^T \Delta\theta + \left( \frac{\lambda^2}{N} \sum_{i=0}^{N} |f_{\theta_t}(x_i) - f_{\theta_t + \Delta\theta}(x_i)|^2 \right)^{1/2} \right) \tag{7}$$

Minimization can be performed in an inner loop by a first order method. We first propose some $\Delta\theta_0 = -\epsilon J = -\epsilon \nabla_\theta C_0$ (for learning rate $\epsilon$) and then iteratively correct this proposal by gradient descent towards $\Delta\theta'$. If only one correction is performed, we can simply add the derivative of the Hilbert-constraining term after $\Delta\theta_0$ has been proposed. However it is possible that solving equation 7 to high precision is beneficial, so we include the possibility of multiple iterations in the algorithm. We found empirically that a single correction was often sufficient.

In the appendix, we tighten the analogy between HCGD and the natural gradient by discussing how one can approximate the natural gradient with an inner first-order optimization loop. We discuss there that evaluating the Fisher and the gradient on the same batch of data causes poor behavior (see also Pascanu & Bengio (2013); Martens (2014)). Failure to use a different batch will result in a type of overfitting, as gradients become, in a sense, judges of their own trustworthiness on the test set. We thus evaluate the empirical expectation of the $L^2$ distance on a validation batch (or, at least, a different batch than used for the initial proposed update). It would also be possible to use unlabeled data. Using a different batch of data for the inner loop ensures that the update does not overfit to the training set at the expense of its previous behavior.

---

**Algorithm 1** Hilbert-constrained gradient descent. Implements Equation 7.

---

**Require:** $n \geq 1$         ▷ Number of corrective steps. May be 1.
**Require:** $\epsilon$         ▷ Overall learning rate
**Require:** $\eta$         ▷ Learning rate for corrective step
**Require:** $\beta$         ▷ Momentum
  1: **procedure**
  2:      $\theta \leftarrow \theta_0$         ▷ Initialize parameters
  3:      $v \leftarrow 0$         ▷ Initialize momentum buffer
  4:      **while** $\theta_t$ not converged **do**
  5:         reset dropout mask, if using
  6:         draw $X \sim \mathbb{P}_x$         ▷ Draw training batch
  7:         $J \leftarrow \nabla_\theta C_0(X)$         ▷ Calculate gradients
  8:         $v \leftarrow \beta v + \epsilon J$
  9:         $\Delta\theta_0 \leftarrow -v$         ▷ Obtain proposed update via SGD with momentum

10:         draw $X_V \sim \mathbb{P}_x$         ▷ Draw validation batch
11:         $g_{L^2} \leftarrow \nabla_{\Delta\theta} \left( \frac{\lambda^2}{N} \sum_{i=0}^{N} |f_{\theta_t}(x_i) - f_{\theta_t + \Delta\theta}(x_i)|^2 \right)^{1/2}$ ▷ Gradient of $L^2$ change. $x_i \in X_V$
12:         $\Delta\theta_1 \leftarrow \Delta\theta_0 - \eta(g_{L^2})$         ▷ First correction towards $\Delta\theta'$
13:         $v \leftarrow v + \eta(g_{L^2})$         ▷ Correct momentum buffer
14:         **for** $1 < j < n$ **do**         ▷ Optional additional corrections
15:           $g_{L^2} \leftarrow J + \nabla_{\Delta\theta} \left( \frac{\lambda^2}{N} \sum_{i=0}^{N} |f_{\theta_t}(x_i) - f_{\theta_t + \Delta\theta_{j-1}}(x_i)|^2 \right)^{1/2}$
16:           $\Delta\theta_j \leftarrow \Delta\theta_{j-1} - \eta(g_{L^2})$
17:           $v \leftarrow v + \eta(g_{L^2})$
18:         $\theta_t \leftarrow \theta_{t-1} + \Delta\theta$

19:      **return** $\theta_t$

---

SGD is commonly improved with momentum. Instead of following the instantaneous gradient $J$, the momentum procedure for SGD follows a 'velocity' term $v$ which is adjusted at each step with

the rule $v \leftarrow \beta v + \epsilon J$. To implement momentum for HCGD, we also keep a velocity term but update it with the final Hilbert-constrained update $\Delta \theta$ rather than $\epsilon J$. The velocity is used to propose the initial $\Delta \theta_0$ in the next update step. We found that this procedure both quickened optimization and lowered generalization error.

Hilbert-constrained gradient descent is computationally cheaper than the exact natural gradient. We are not required to approximately invert any large matrix $F$, nor are we required to calculate the any per-example gradients separately. When the validation batch $X_V$ is drawn anew for each corrective iteration (step 8 in Algorithm 2), HCGD requires an additional two forward passes and one backwards pass for each correction $i < n$, for a total of $2 + 3n$ passes each outer step. This can be reduced by 1 pass for $i \geq 1$ if $X_V$ is drawn anew just for $i = 0$.

## 3 RESULTS

We demonstrate our method on training networks at the task of MNIST digit classification, with two network architectures, and on image classification in the CIFAR10 dataset. In all tests, we use a tuned learning rate $\epsilon$ for SGD, and then use the same learning rate for HCGD. We use values of $\lambda = 0.5$ and $\eta = 0.02$. (For the $n = 1$ version, $\lambda$ can be folded into the inner learning rate $\eta$. Values were chosen so that $\lambda \eta = 0.01$.) We chose the batch size for the "validation" batch to be 256. While the examples in each "validation" batch were different than the training batch, they were also drawn from the train set. All models were implemented in PyTorch (Paszke et al. (2017)).

We focus first on the clean example of training a dense multilayer perceptron without any modifications. We employ an $850 - 90 - 50 - 10$ architecture with ReLU activations, and do not use dropout or batch normalization. The output is a softmax activation and the cost is the cross-entropy. As can be seen in Figure 1, HCGD notably improves performance on the test set. This is ture for both algorithms with momentum (1b) and without momentum (1c). We use $\epsilon = 0.04$ with momentum and $\epsilon = 0.1$ without momentum. The versions in which the gradient is corrected towards the ideal Hilbert-constrained update just once ($n = 1$) or many times ($n = 10$) behaved similarly. We use only the $n = 1$ version in future tests.

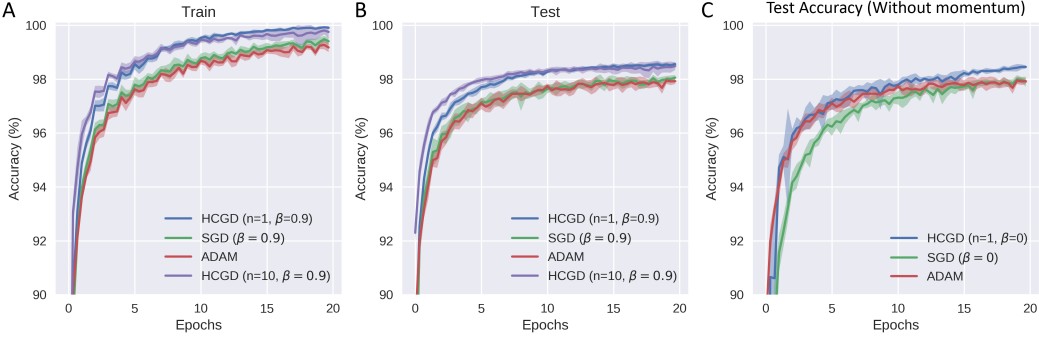

Figure 1: Test and train accuracy on MNIST digit classification for a dense multilayer perceptron without dropout or normalization. Traces and envelopes represent the mean $\pm$ standard deviation of the traces of 10 runs. HCGD converges faster than SGD and generalizes better both for $n = 1$ and $n = 10$ corrections to the gradient. Both SGD and HCGD employ momentum ($\beta$) in (a) and (b), but use no momentum in (c).

While HCGD converges more quickly than SGD, it requires a larger number of passes through the computational graph. In Figure 2, we plot the same results as Figure 1a,b but choose the total number of passes as the $x$-axis. It can be seen that when the number of inner-loop corrections $n = 1$, HCGD converges in similar compute time as SGD (but generalizes better).

Next, we test the various optimizers at training a 3-layer convolutional neural network with Batch Normalization (BN) on MNIST digit classification. HCGD still outperforms standard SGD and ADAM, but by a much slimmer margin (Figure 3). Batch normalization has the effect that the activations, and thus the gradients, become normalized and whitened. For a BN network, then,

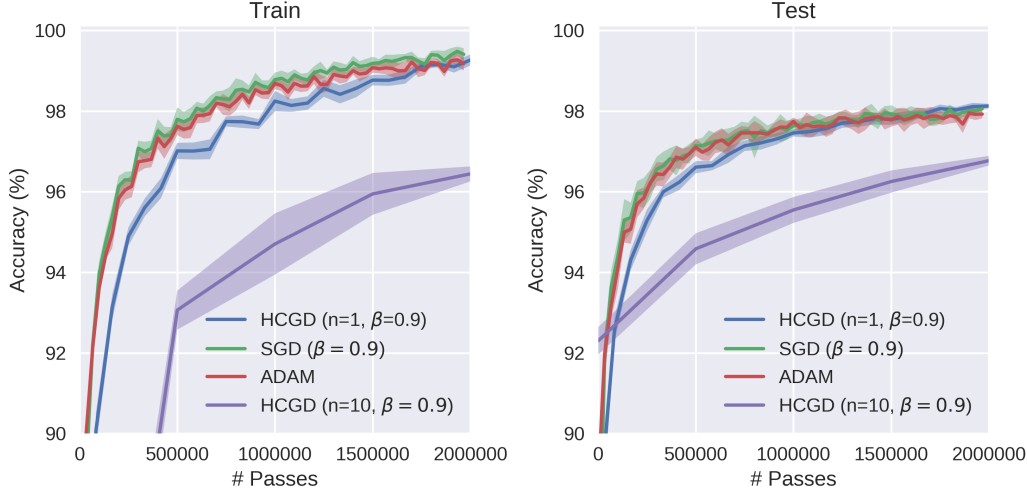

Figure 2: Test and train accuracy on MNIST digit classification for a dense multilayer perceptron, plotted against the total passes performed (where total passes = inference passes + backprop passes). The x axis has been truncated at the run time of SGD. Traces and envelopes represent the mean ± standard deviation of the traces of 10 runs.

SGD becomes more similar to the natural gradient. This may explain why HCGD offers a smaller improvement. Both the natural gradient and HCGD regularize the change in the outputs throughout learning, just by a different metric. Networks with BN are thus already regularized in a manner similar to HCGD, which leads to a slimmer advantage.

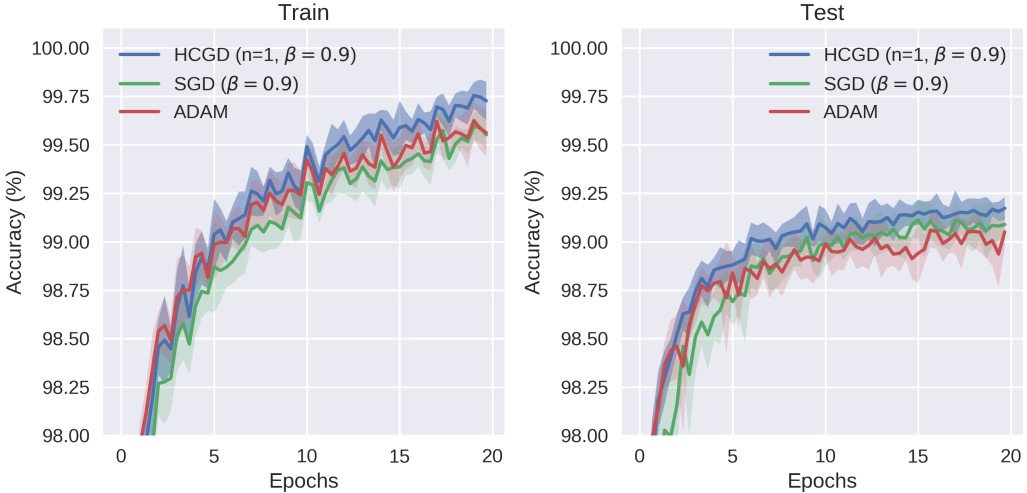

Figure 3: Test and train accuracy on MNIST digit classification for a convolutional neural network trained with Batch Normalization. Traces and envelopes represent the mean ± standard deviation of the traces of 10 runs.

Finally, we test the performance of HCGD as applied to the CIFAR10 image classification problem. We train a Squeezenet v1.1, a convolutional neural network model optimized for parameter efficiency(Iandola et al., 2016). As is common for training large models, we train at a large learning rate and then decrease the learning rate by a factor of 10 for fine tuning. When HCGD is trained with the same learning rate as SGD (initial $\epsilon = 0.1$), it outperforms SGD while the learning rate is

high, but performs worse than SGD once the learning rate is decreased (Figure 5). We suspect that this is because HCGD effectively reduces the learning rate, removing some of the positive annealing effects of a high initial learning rate. Note that we also decrease the inner learning rate $\eta$ by a factor of 10. When we increase the initial learning rate such that the test error of HCGD matches that of SGD, the final train error decreases below that of SGD. Averaging the final 40 epochs to reduce noise, SGD achieves an mean error percentage of 8.00.01, while HCGD at $\epsilon = 0.3$ achieved an error of 7.80.025 (with  indicating the standard error of the mean). HCGD thus generally decreases the test error at a given learning rate, but needs to be trained at a higher learning rate to a achieve a given level of gradient noise.

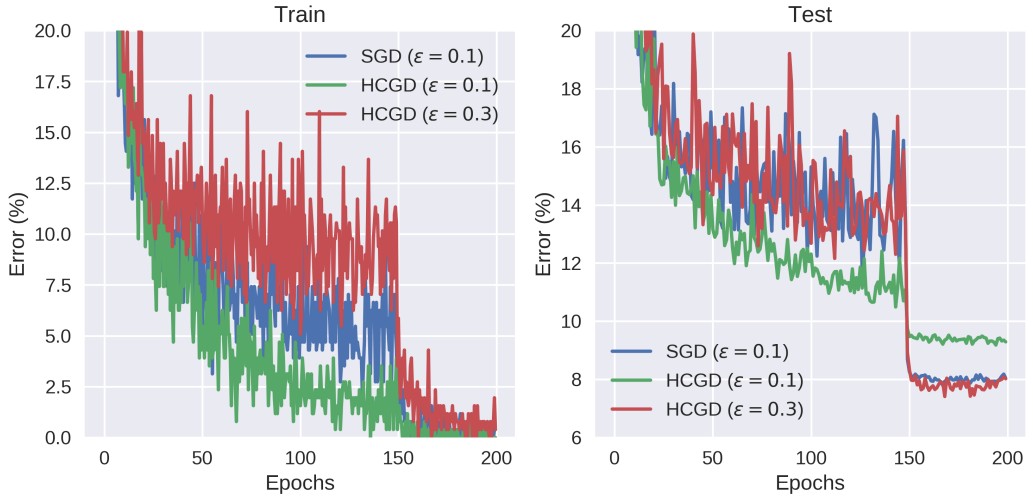

Figure 4: Results of a Squeezenet v1.1 trained on CIFAR10. The learning rate $\epsilon$ is decreased by a factor of 10 at epoch 150. HCGD requires a higher learning rate to a achieve a similar level of gradient noise, which is important to decrease overfitting.

## 4 THE IMPORTANCE OF CONSIDERING BEHAVIOR IN FUNCTION SPACE

A central theme of this work is that regularization and optimization should occur in the space of functions, not merely the space of parameters. In this section we investigate this difference.

In the space of parameters, SGD is a strongly local update rule. Large jumps are generally prohibited. SGD is thus more likely to find solutions that are close to the initialization, and furthermore to trace a path of limited length. This discourages the sampling a large volume of parameter space during optimization, which may lead to overfitting. This locality may partly explain the unexpected generalization abilities of SGD (e.g. Zhang et al. (2016)). Early stopping, which lowers generalization error, also limits great exploration.

If SGD is successful due to its movement though parameter space, then it moves similarly in function space only to the extent to which there is a reasonably smooth mapping between these two spaces. Distances in these two spaces may still be qualitatively different.

We can partially examine the difference between parameter and function space by examining how a network moves differently through them. In figure 5, we have plotted the cumulative squared distance traveled during optimization for an example network trained on MNIST. Figure 5a,c display the cumulative squared distance traveled (CSDT) in $L^2$ function space, while Figure 5b,d display the CSDT in parameter space. We first examine the behavior of SGD. While SGD moves more more slowly over time through parameter space (5c), reflecting a decreasing gradient, the CSDT of SGD in $L^2$-space grows linearly for a large portion of optimization. Ideally, a network would cease to change before entering the overfitting regime. However, the network continues to drift through $L^2$-space even after the test error saturates at around epoch 15 (see Figure 1). Note that SGD with

momentum differs from plain SGD in that the scale of the total distance traveled is significantly reduced (Figure 5b,d). This effect could be due to a decreased level of noise in the update. Network drift in parameter space and in $L^2$-space thus display qualitatively different behavior for SGD.

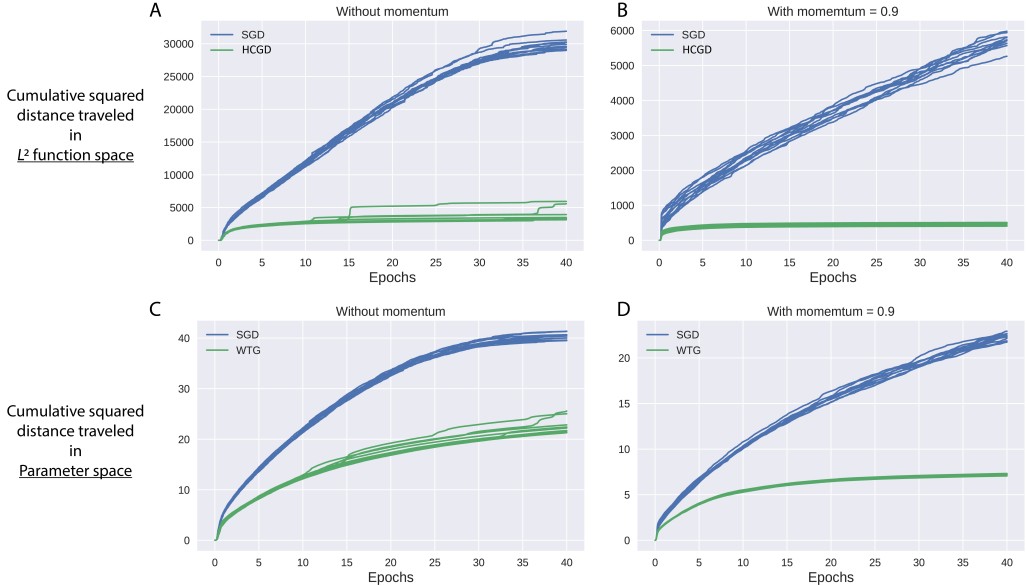

Figure 5: The cumulative squared distance traveled through $L^2$-space, top row, and through parameter space, bottom row, for an MLP trained on MNIST. It can be seen that SGD continues to drift in $L^2$-space during the overfitting regime, while HCGD plateaus. This is true for networks trained with momentum, left column, and without momentum, right column. Note that momentum significantly decreases the scale of distance traveled. Individual traces represent random seeds. We measure distance in $L^2$-space in the same manner as in the HCGD algorithm: by registering the Euclidean distance between the network's outputs on a single validation batch before and after an update.

The HCGD algorithm is designed to reduce motion through $L^2$-space. Figure 5a,b show that HCGD indeed greatly reduces motion through $L^2$-space whether or not momentum is used. The plateau of the CSDT indicates that the function has converged to a single location; it ceases to change. SGD, on the other hand, does not converge to a single function even long after test error saturates. It is interesting to note that HCGD allows the parameters to continue to drift (5c,d) even though the function has generally converged.

## 5 DISCUSSION

Neural networks encode functions, and it is important to consider the behavior of optimizers through the space of possible functions. The $L^2$ Hilbert space defined over distribution of input examples is a tractable and useful space for analysis. In this paper we propose to regularize the change in $L^2$ space between successive updates. The idea is to limit the movement of the function, just as gradient descent limits movement of the parameters. Our resulting learning rule, Hilbert-constrained gradient descent (HCGD), increases test performance on standard image classification architectures. We hope that this work inspires more thought and analysis of behavior in $L^2$-space.

A alternative explanation of our algorithm is that it penalizes directions that are very sensitive controls of the outputs, similar to the natural gradient, while still allowing learning. In addition, since we evaluate the change in $L^2$-space and the gradient on different data, HCGD asks the model to learn from current examples only in ways that will not affect what has already been learned from other examples. These intuitions are equivalent to the idea of limiting changes in $L^2$-space.

Given these empirical results, it would be desirable to theoretically prove better generalization bounds for a method regularized in $L^2$-space. One promising framework is stability analysis, which

has recently been applied to establish some bounds on the generalization error of SGD itself (Hardt et al. (2015)). It can be shown that generalization error is bounded by the stability of an algorithm, defined as the expected difference of the loss when two networks are trained on datasets that are identical except for one example. Hardt et al. (2015) analyzes the stability of SGD in parameter space, then uses a Lipschitz condition to move to function space and bound the stability of the error. We expect that bounding the movement through $L^2$-space leads to increased error stability compared to bounding movement through parameter space (as is done by SGD), simply by removing reliance on the assumed Lipschitz condition. We leave a proof of this idea to later work.

It interesting to ask if there is support in neuroscience for learning rules that diminish the size of changes when that change would have a large effect on other tasks. It is unlikely that the nervous system performs precisely the natural gradient or HCGD, but there is some evidence that some analog is in play. One otherwise perplexing finding is that behavioral learning rates in motor tasks are dependent on the direction of an error but independent of the magnitude of that error (Fine & Thoroughman (2006)). This result is not expected by most models of gradient descent, but would be expected if the size of the change in the output distribution (i.e. behavior) were regulated to be constant. Regularization upon behavior change (rather than synaptic change) would predict that neurons that are central to many actions, like neurons in motor pools of the spinal cord, would learn very slowly after early development, despite the fact that their gradient to the error on any one task (if indeed it is calculated) is likely to be quite large. Given our general resistance to overfitting during learning, and the great variety of roles of neurons, it is likely that some type of regularization of behavioral and perceptual change is at play.

## ACKNOWLEDGMENTS

The authors would like to thank helpful discussions with Roozbeh Farhoodi and David Rolnick. A.B. and K.K. thank NIH grants R01NS063399, R01NS074044, and MH103910.

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

## A   NATURAL GRADIENT BY GRADIENT DESCENT

In order to better compare the natural gradient to the Hilbert-constrained gradient, we propose a natural gradient algorithm of a similar style.

Previous work on the natural gradient has aimed to approximate $F^{-1}$ as best and as cheaply as possible. This is equivalent to minimizing Equation 2 (i.e. $J\Delta\theta + \frac{\lambda}{2}\Delta\theta^T F\Delta\theta$) with a single iteration of a second-order optimizer. For very large neural networks, however, it is much cheaper to calculate matrix-vector products than to approximately invert a large matrix. It is possible that the natural gradient may be more accessible via an inner gradient descent, which would be performed during each update step as an inner loop.

We describe this idea at high level in Algorithm 2. After an update step is proposed by a standard optimizer, the algorithm iteratively corrects this update step towards the natural gradient. To start with a good initial proposed update, it is better to use a fast diagonal approximation of the natural gradient (such as Adagrad or RMSprop) as the main optimizer. Each additional correction requires just one matrix-vector product after the gradients are calculated. Depending on the quality of the proposed update, the number of iterations required is likely to be small, and even a small number of iterations will improve the update.

---

**Algorithm 2** Natural gradient by gradient descent. This algorithm can be paired with any optimizer to increase its similarity to the natural gradient.

---

**Require:** $n$                    ▷ Number of corrective steps. May be 1.
**Require:** $\eta$                        ▷ Learning rate for corrective step
 1: **procedure**
 2:      $\theta \leftarrow \theta_0$                   ▷ Initialize parameters
 3:      **while** $\theta_t$ not converged **do**
 4:          $\Delta\theta_0 \leftarrow \text{RMSprop}(\theta_t)$      ▷ Use any optimizer to get proposed update
 5:          **for** $i < n$ **do**                 ▷ Corrective loop
 6:             $\Delta\theta_{i+1} = \Delta\theta_i - \eta(J + \lambda F\Delta\theta_i)$      ▷ Step towards $\frac{1}{\lambda}F^{-1}J$
 7:          $\theta \leftarrow \theta + \Delta\theta$
 8:      **return** $\theta_t$

---

Since the Fisher matrix $F$ can be calculated from the covariance of gradients, it never needs to be fully stored. Instead, for an array of gradients $G$ of size (# parameters, # examples), we can write

$$F\Delta\theta = (GG^T)\Delta\theta = G(G^T\Delta\theta) \tag{8}$$

The choice of $G$ is an important one. It cannot be a vector of aggregated gradients (i.e. $J$), as that would destroy covariance structure and would result in a rank-1 Fisher matrix. Thus, we must calculate the gradients on a per-example basis. To compute $G$ efficiently it is required that a deep learning framework implement forward-mode differentiation, which is currently not supported in popular frameworks.

If we choose $G$ to be the array of per-example gradients on the minibatch, $F$ is known as the 'empirical Fisher'. As explained in Martens (2014) and in Pascanu & Bengio (2013), the proper method is to calculate G from the predictive (output) distribution of the network, $\mathbb{P}_\theta(y|x)$. This can be done as in Martens & Grosse (2015) by sampling randomly from the output distribution and re-running backpropagation on these fictitious targets, using (by necessity) the activations from the minibatch. Alternatively, as done in Pascanu & Bengio (2013), one may also use unlabeled or validation data to calculate $G$ on each batch.

