# OpenReview forum: "Improving generalization by regularizing in $L^2$ function space"
_ICLR.cc/2018/Conference — Reject_

### Official Review · AnonReviewer3 · 2017-11-13
**improved during review**

**Rating:** 6
**Confidence:** 3

**Review:**


GENERAL IMPRESSION:

Overall, the revised version of the paper is greatly improved. The new derivation of the method yields a much simpler interpretation, although the relation to the natural gradient remains weak (see below). The experimental evaluation is now far more solid. Multiple data sets and network architectures are tested, and equally important, the effect of parameter settings is investigated. I enjoyed the investigation of the effect of L_2 regularization on qualitative optimization behavior.


CRITICISM:

My central criticism is that the introduction of the L_2 norm as a replacement of KL divergence is completely ad-hoc; how it is related to KL divergence remains unclear. It seems that other choices are equally well justified, including the L_2 norm in parameter space, which then defeats the central argument of the paper. I do believe that L_2 distance is more natural in function space than in parameter space, but I am missing a strict argument for this in the paper.

Although related work is discussed in detail in section 1, it remains unclear how exactly the proposed algorithm overlaps with existing approaches. I am confident that it is easy to identify many precursors in the optimization literature, but I am not an expert on this. It would be of particular interest to highlight connections to algorithm regularly applied to neural network training. Adadelta, RMSprop, and ADAM are mentioned explicitly, but what exactly are differences and similarities?

The interpretation of figure 2 is off. It is deduced that HCGD generalizes better, however, this is the case only at the very end of training, while SGD with momentum and ADAM work far better initially. With the same plot one could sell SGD as the superior algorithm. Overall, also in the light of figure 4, the interpretation that the new algorithm results in better generalization seems to stand on shaky ground, since differences are small.

I like the experiment presented in figure 5 in particular. It adds insights that are of value even if the method should turn out to have significant overlap with existing work (see above), and perform "only" on par with these: it adds an interesting perspective to the discussion of how network optimization "works", how it handles local optima and which role they play, and how the objective function landscape is "perceived" by different optimizers. This is where I learned something new.


MINOR POINTS:

page 5: "the any" (typo)

page 5: "ture" -> "true" (typo)

---

### Official Review · AnonReviewer2 · 2017-11-16
**Ideas based on WGAN but for NN training; some experiments show some improvement but they are not complete/convincing. Rejection**

**Rating:** 5
**Confidence:** 4

**Review:**

I have read comments and rebuttal - i do not have the luxury of time to read in depth the revision.
It seems that the authors have made an effort to accommodate reviewers' comments. I upgraded the rating.

-----------------------------------------------------------------------------------------------------------------------

Summary: The paper considers the use of natural gradients for learning. The added twist is the substitution of the KL divergence with the Wasserstein distance, as proposed in GAN training. The authors suggest that Wasserstein regularization improves generalization over SGD with a little extra cost.

The paper is structured as follows:
1. KL divergence is used as a similarity measure between two distributions.
2. Regularizing the objective with KL div. seems promising, but expensive.
3. We usually approximate the KL div. with its 2nd order approximation - this introduces the Hessian of the KL divergence, known as Fisher information matrix.
4. However, computing and inverting the Fisher information matrix is computationally expensive.
5. One solution is to approximate the solution F^{-1} J using gradient descent. However, still we need to calculate F. There are options where F could be formed as the outer product of a collection gradients of individual examples ('empirical Fisher').
6. This paper does not move towards Fisher information, but towards Wasserstein distance: after a "good" initialization via SGD is obtained, the inner loop continues updating that point using the Wasserstein regularized objective.
7. No large matrices need to be formed or inverted, however more passes needed per outer step.

Importance:
Somewhat lack of originality and poor experiments lead to low importance.

Clarity:
The paper needs major revision w.r.t. presenting and highlighting the new main points. E.g., one needs to get to page 5 to understand that the paper is just based on the WGAN ideas in Arjovsky et al., but with a different application (not GANS).

Originality/Novelty:
The paper, based on WGAN motivation, proposes Wasserstein distance regularization over KL div. regularization for training of simple models, such as neural networks. Beyond this, the paper does not provide any futher original idea. So, slight to no novelty.

Main comments:
1. Would the approximation of C_0 by its second-order Taylor expansion (that also introduces a Hessian) help? This would require the combination of two Hessian matrices.

2. Experiments are really demotivating: it is not clear whether using plain SGD or the proposed method leads to better results.

Overall:
Rejection.

---

### Official Review · AnonReviewer1 · 2017-11-28
**The paper has a weak theoretical justification and poor choice of baseline method for empirical evaluation.**

**Rating:** 4
**Confidence:** 3

**Review:**

The paper presents an additive regularization scheme to encourage parameter updates that lead to small changes in prediction (i.e. adjusting updates based on their size in the output space instead of the input space). This goal is to achieve a similar effect to that of natural gradient, but with lighter computation. The authors claim that their regularization is related to Wasserstein metric (but the connection is not clear to me, read below). Experiments on MNIST with show improved generalization (but the baseline is chosen poorly, read below).

The paper is easy to read and organized very well, and has adequate literature review. However, the contribution of the paper itself needs to be strengthened in both the theory and empirical sides.

On the theory side, the authors claim that their regularization is based on Wasserstein metric (in the title of the paper as well as section 2.2). However, this connection is not very clear to me [if there is a rigorous connection, please elaborate]. From what I understand, the authors argue that their proposed loss+regularization is equivalent to the Kantorovich-Rubinstein form. However, in the latter, the optimization objective is the f itself (sup E[f_1]-E[f_2]) but in your scheme you propose adding the regularization term (which can be added to any objective function, and then the whole form loses its connection to Wasserstrin metric).

On the practical side, the chosen baseline is very poor. The authors only experiment with MNIST dataset. The baseline model lacks both "batch normalization" and "dropout", which I guess is because otherwise the proposed method would under-perform against the baseline. It is hard to tell if the proposed regularization scheme is something significant under such poorly chosen baseline.

---

### Author Response · Authors · 2018-01-05
**We have significantly reworked this project and submitted the revised version**

Based on the reviewers' responses, we have opted for a major rewrite of this project. The major changes include a change of theoretical justification, empirical analyses of this justification, additional optimization experiments, and a rework of the text (including the title and abstract).

1.
The largest change is a shift away from the framework of Wasserstein distance in favor of one that establishes a metric of function distance in an L^2 Hilbert space, and regularizes that distance throughout learning. We believe this framework better emphasizes our overall message, which is that it is important to consider the behavior of optimizers in function space, not just parameter space.

As reviewer 1 noted, our original claim of regularizing the Wasserstein distance of the output distribution was not quite correct. We were limiting the change on specific examples, rather than between two distributions. After some thought, we realized that the expected norm of the change in output on specific examples is equivalent to an L^2 distance in a Hilbert space. A better description of our algorithm, then, is that it performs gradient descent through this L^2 function space.

This conceptual change required that we re-name the learning rule. The title, abstract, and algorithm description reflect this change.

2.
The foundation of this work is a shift from thinking in parameter space to thinking in functional space. Previously, we did not adequately highlight or discuss this shift. We included a set of plots designed to evaluate the typical motion of a network through both parameter space and function space. These plots serve to justify an L^2 function space as a useful one for analysis.

3.
All three reviewers were unsatisfied with the small number of optimization experiments. We included more experiments, including those using techniques like momentum, batch normalization, and alternative architectures.

---

### Decision · Program_Chairs · 2018-01-29
**ICLR 2018 Conference Acceptance Decision**

**Decision:**

Reject

**Comment:**

Dear authors,

Despite the desirable goal, that is to move away from regularization in parameter space toward regularization in function space, the reviewers all thought that the paper was not convincing enough, both in the choice of the particular regularization and in the experimental section.

While I appreciate that you have done a major rework of the paper, the rebuttal period should not be used for that and we can not expect the reviewers to do a complete re-review of a new version.

This paper thus cannot be accepted to ICLR.